# Mitochondrial Dynamics in Stem Cells and Differentiation

**DOI:** 10.3390/ijms19123893

**Published:** 2018-12-05

**Authors:** Bong Jong Seo, Sang Hoon Yoon, Jeong Tae Do

**Affiliations:** Department of Stem Cell and Regenerative Biotechnology, Konkuk Institute of Technology, Konkuk University, Seoul 05020, Korea; sbj1990@naver.com (B.J.S.); kei99137@naver.com (S.H.Y.)

**Keywords:** mitochondria, mitochondrial dynamics, fusion, fission, pluripotency, differentiation

## Abstract

Mitochondria are highly dynamic organelles that continuously change their shape. Their main function is adenosine triphosphate (ATP) production; however, they are additionally involved in a variety of cellular phenomena, such as apoptosis, cell cycle, proliferation, differentiation, reprogramming, and aging. The change in mitochondrial morphology is closely related to the functionality of mitochondria. Normal mitochondrial dynamics are critical for cellular function, embryonic development, and tissue formation. Thus, defects in proteins involved in mitochondrial dynamics that control mitochondrial fusion and fission can affect cellular differentiation, proliferation, cellular reprogramming, and aging. Here, we review the processes and proteins involved in mitochondrial dynamics and their various associated cellular phenomena.

## 1. Introduction

Mitochondria are cytoplasmic organelles of cells and function as energy stations for adenosine triphosphate (ATP) production. The major functions of mitochondria are aerobic energy production, reactive oxygen species (ROS) production, calcium homeostasis, cellular signaling pathways, and synthesis and/or assembly of cellular metabolites, such as fatty acids, amino acids, iron/sulfur clusters, pyrimidines, heme, and steroid hormones [1,2,3]. Mitochondrial dysfunction causes aging, loss of synaptic nerve cells, and cell death in many human neurological diseases [4,5]. The shape of a mitochondrion is directly or indirectly determined by several factors. The indirect determinants of mitochondrial shape include several environmental conditions, such as a low-oxygen [6], and a high demand for energy [7] and metabolites [8]. The shape of mitochondria is also directly regulated by mitochondrial intermembrane proteins and their accessory proteins [9]. Numerous researchers have studied the role of these proteins in various cell types in determining the shape of mitochondria. However, the results of these studies are rather uninformative and lack an understanding of the underlying mechanisms, especially in stem cells. In this review, we describe the basic mechanisms of functioning of the proteins involved in mitochondrial dynamics, without details on the relative mechanisms. Furthermore, we focus on how these proteins affect cellular metabolism, reprogramming, differentiation, and aging.

## 2. Components Determining the Mitochondrial Structure

Mitochondria are present in most cell types and tissues, and mitochondrial shape is changed exquisitely by the process of fusion and fission. The mitochondrial movement was observed under an optical microscope nearly 100 years ago [10]. This process is involved in the growth and division of mitochondria and is important in maintaining the number and functions of mitochondria [11]. 

In response to the metabolic changes, mitochondrial morphology is changed through both fission and fusion. Mitochondrial fission is essential for growing and dividing cells to populate them with sufficient numbers of mitochondria [12]. Mitochondrial fusion occurs more often in cells that rely on oxidative phosphorylation [13]. Mouse embryonic fibroblasts (MEFs) defective in mitochondrial fusion could survive, but some of their mitochondria displayed a reduced mitochondrial DNA (mtDNA) copy number, a loss of membrane potential, and problems with adenosine triphosphate (ATP) synthesis [14]. Mitochondrial fusion is not absolutely essential for cell survival, but it is essential for normal development [15]. 

### 2.1. Post-Translational Modification of Mitochondrial Proteins

Mitochondrial fission is regulated by post-translational modification of the dynamin-related protein 1 (Drp1) protein, including modification by phosphorylation, *S*-nitrosylation, small ubiquitin-like modifier (SUMO)-ylation, ubiquitination, and *O*-GlcNAc modification (*O*-GlcNAcylation), in response to a variety of cellular stimuli [12]. Mitochondrial fusion is a two-step process involving the outer mitochondrial membrane (OMM) fusion, mediated by mitofusin proteins (Mfn1 and Mfn2) [16,17,18,19,20], and inner mitochondrial membrane (IMM) fusion, mediated by Opa1, and could possibly be coupled [21,22,23]. We describe here the functions of proteins involved in mitochondrial fusion and fission (Figure 1).

Protein phosphorylation is a posttranslational modification that regulates protein function and signaling pathways [24]. The phosphorylation site in Ser637 of the Drp1 protein, which is a highly conserved consensus region among metazoans, is important for mitochondrial division [25,26,27]. Modification of this site by the inhibition of GTPase activity results in the inhibition of mitochondrial fission [25,26,27]. 

*S*-nitrosylation also regulates the function of Drp1 through a specific modification. This is a redox-related modification of thiols at the cysteine residue by nitric oxide (NO) that affects various proteins involved in a number of cellular processes [28]. NO functions as a signaling molecule, and it can cause neuronal damage, in part by excessive mitochondrial fragmentation [29]. Thus, *S*-nitrosylation can be a tool for modulating Drp1 function.

The small ubiquitin-like modifier (SUMO) protein also participates in Drp1 modification [30]. SUMO attachment to a protein often changes the subcellular localization of proteins or prevents ubiquitin-mediated degradation. Especially, SUMOylation of Drp1 increases Drp1 protein levels and mitochondrial fission [31]. Braschi et al. suggested that mitochondrial-anchored protein ligase (MAPL) was the SUMO E3 ligase targeting Drp1 proteins [32,33]. Drp1 is also modified by ubiquitination. Membrane-bound E3 ubiquitin ligases, such as MARCH-V, MARCH3, and MITOL, are involved in the mitochondrial dynamics by reducing the level of mitochondrial fission proteins [34,35,36].

Addition of O-linked *N*-acetylglycosamine (*O*-GlcNAc) was suggested to be involved in the modification of mitochondrial proteins, and *O*-GlcNAcylation of mitochondrial proteins showed negative effects on mitochondrial function [37]. Increased mitochondrial *O*-GlcNAcylation levels were involved in mitochondrial dysfunction in the cariac myocyte [38]. However, reduced levels of global *O*-GlcNAcylation were correlated with mitochondrial dysfuction and disruption of the mitochondrial network, and has been observed in the brains of patients with Alzheimer’s disease [39]. Thus, the effect of *O*-GlcNAcylation of mitochondrial proteins needs to be further clarified. 

### 2.2. Mitochondrial Fission Proteins

Mitochondrial division in a cell contributes to ensuring proper distribution and quality control of mitochondria, which maintain a cell in a healthy state. A member of the dynamin family of GTPases, dynamin-like 1 (Dnm1), which is also referred to as Drp1, is a major player of mitochondrial division [40,41,42]. Genetic and biochemical studies in yeast have shown that Dnm1-mediated mitochondrial cleavage requires the tail-anchored OMM protein, Fis1, and an adaptor protein, Mdv1, or its paralogue, Caf4, which connect Dnm1 to Fis1 [11,43,44,45,46,47,48,49,50]. During mitochondrial division, Fis1 transiently interacts with cytosolic Dnm1 by the tetratricopeptide-repeat motif via the cytosolic adapter protein, Mdv1/Caf4, indicating that Fis1 functions as a mitochondrial Dnm1 receptor [46]. However, Mdv1 and its homologs (Caf4, Num1, and Mdm36) were not found in mammalian cells, indicating that only two proteins, Dnm1 and Fis1, are conserved in all species that contain mitochondria [51]. Several fission-related proteins have been identified in mammals, but their detailed mechanistic role in mitochondrial fission has not been clarified. Several reports showed that endoplasmic reticulum (ER) mitochondria connections contribute to the mitochondrial fission process [52,53,54]. Csordas et al., using an electron microscope (EM), reported that the mitochondria and ER have a connection [52]. This particular structure, called ER mitochondria encounter structure (ERMES), has been implicated in mitochondrial fission before Drp1 is recruited to the mitochondria [53]. Therefore, the formation of ERMES possibly plays a crucial role in mitochondrial fission. Indeed, Korobova et al. showed that the ER-bound protein, inverted formin-2 (INF2), promotes ER contractions around the mitochondria in the early stages of mitochondrial fission [54].

### 2.3. Mitochondrial Fission Accessory Proteins

Besides the major mitochondrial fission proteins, mitochondrial fission accessory proteins, such as mitochondrial fission factor (Mff), mitochondrial dynamics 49 (MiD49), mitochondrial dynamics 51 (MiD51), ganglioside-induced differentiation-associated protein 1 (GDAP1), and endophilins, additionally play crucial roles in mitochondrial fission.

Mff is a C-terminal-tail immobilized protein identified in a *Drosophila* RNA interference (RNAi) library used to search for mitochondrial morphological changes. Mammalian mitochondria also contain an orthologue of Mff, suggesting that Mff may be involved in the mitochondrial division and fission in mammalian cells [55]. Mff overexpression caused mitochondrial fragmentation, similar to Drp1 overexpression in mammalian cells [55,56,57]. Consistent with these observations, in vitro and in vivo experiments have demonstrated that Mff transiently interacts with Drp1 through the N-terminal cytoplasmic domain.

MiD51 and MiD49 variants, known as mitochondrial elongation factor 1 and 2 (MIEF1/2), respectively, are OMM proteins identified by random cell localization screens of raw proteins that cause unique distribution and changes in mitochondrial morphology [58]. MIEF1/2 form foci and rings around mitochondria and directly recruit cytosolic Drp1 to the mitochondrial outer membrane surface [59], serving as adaptors linking Drp1 and Mff [58]. Therefore, MIEF1/2 was suggested to be a receptor for Drp1 and a mediator of mitochondrial division (fission). MIEF1/2 knockdown by RNAi resulted in the reduction of the interaction of Drp1 with mitochondria, leading to mitochondrial elongation. Surprisingly, overexpression of MIEF1/2 induced mitochondrial fission by sequestering Drp1 protein activity [58,59]. Zhao et al., on the other hand, claimed that the knockdown of MIEF1 by RNAi induces mitochondrial fragmentation. They concluded that MIEF1 functions as a Drp1 suppressor that inhibits GTPase-dependent fission activity of Drp1 and MIEF1 also has a role independent of Mfn2 in the fusion pathway [60]. Given the discrepancy, more research concerning MIEF1/2 must be carried out. 

GDAP1 is another mitochondrial division-related factor located on the OMM through the C-terminal hydrophobic transmembrane domain, which pushes the bulk N-terminal domain to the cytoplasm [61]. It is expressed in myelinating Schwann cells and motor and sensory neurons [62]. The GDAP1 mutation induced progression to peripheral nerve injury Charcot-Marie-Tooth disease, with primary axonal damage and primary dehydration of the peripheral nerve [63]. GDAP1 mutants found in patients with the Charcot-Marie-Tooth disease do not target mitochondria and lack mitochondrial cleavage activity [64]. GDAP1-induced mitochondrial fragmentation was inhibited by Drp1 knockdown or the expression of a dominant-negative Drp1-K38A mutation, indicating that GDAP1 is a Drp1-dependent modulator of mitochondrial division [65].

Endophilins, fatty acyl transferases, were proposed to mediate membrane curvature changes and participate in membrane cleavage during endocytosis and intracellular organelle biogenesis [66]. They have an N-terminal Bar domain interacting with the membrane and a C-terminal SH3 domain mediating protein binding [67,68,69,70]. Endophilin B1 (also called Endo B1, Bif-1) was identified by a yeast two-hybrid protein screen to bind to Bax, a proapoptotic Bcl-2 family member, and was reported to be involved in apoptosis, mitochondrial morphogenesis, and autophagosome formation [71,72,73,74]. 

### 2.4. Mitochondrial Fusion Proteins

At the molecular level, mitochondrial fusion is a two-step process that requires coordinated sequential fusion of the OMM and IMM [75,76,77]. In mammals, this process relies on the unique mitochondrial sub-localization of the three fusion-related proteins: The OMM-located mitofusin 1 and 2 (Mfn1 and Mfn2) and IMM-located optic atrophy 1 (Opa1) [19,78].

The mitofusin proteins, Mfn1 and Mfn2, belong to the ubiquitous transmembrane GTPase family, which is conserved from yeast to human [79,80]. Mfn1 and Mfn2 share about 80% genomic sequence similarity and show the same structural motifs [18,20]. Their amino terminal GTPase domain contains five motifs, each of which plays an important role in GTP binding and hydrolysis [81]. Notably, the proline-rich region (PR) involved in protein-protein interactions is found only in Mfn2. Mfn1 and Mfn2 double-knockout (DKO) mice die prematurely during pregnancy due to insufficient mitochondrial fusion in the placenta [20,82]. Interestingly, double-mutant embryos die without any visible developmental defect, suggesting the non-redundant function of Mfn1 and Mfn2 in embryonic development. Indeed, Mfn1 mediates mitochondrial docking and fusion more efficiently than Mfn2, presumably due to its high GTPase activity [83]. Furthermore, Mfnl is required to mediate Opa1-induced mitochondrial fusion, but not Mfn2 [22].

Opa1 is also a dynamin family GTPase that promotes IMM fusion following OMM fusion [21,84]. Cryo-immunogold EM analysis revealed that Opa1 is a mitochondrial intermembrane space protein [85]. The Opa1 function is controlled in part by proteolysis, by which Opa1 is cleaved and mitochondrial fusion is blocked [86,87]. Proteolytic inactivation of Opa1 could induce the change of mitochondrial morphology, such as swelling and constriction of mitochondrial tubules and swollen cristae [85]. In addition, Opa1 was suggested to help maintain cristae morphology, like Mitofilin and ATP synthase [88]. As cristae shape is important for the assembly of respiratory chain complexes and respiratory efficiency, Opa1 may be essential for the proper assembly and function of the electron transport supercomplex [23,89].

### 2.5. Mitochondrial Fusion Accessory Proteins

Besides the three major mitochondrial fusion proteins, some accessory proteins, such as PINK1 (protein phosphatase and tensin homolog (PTEN)-induced kinase 1) and PARKIN, could affect the mitochondrial fusion machinery. PINK1 is a ubiquitin kinase that phosphorylates ubiquitin and subsequently activates the ubiquitin ligase, PARKIN. PINK1 and PARKIN have been suggested as inducing factors for mitophagy. When PINK1 is stabilized on the OMM of malfunctioning mitochondria, PINK1 recruits ubiquitin E3 ligase kinase and autophagy receptors, which leads to autophagosome biogenesis and subsequent catabolism by lysosomes [90]. PINK1 has little activity in normal mitochondria [91]. However, when depolarization occurs in the mitochondria, the destabilization process of PINK1 is stopped and PINK1 accumulates and phosphorylates the substrate proteins [92]. Healthy mitochondria actively degrade PINK1 to prevent mitophagic destruction. However, damaged mitochondria no longer trigger PINK1 degradation, resulting in the accumulation of PINK1 in mitochondria followed by the mitophagic destruction of the organelle [93].

Parkinson’s disease can be caused by a mutation in Pink1 or Parkin, which may lead to the accumulation of damaged mitochondria in neurons. Ultimately, damaged mitochondria in patients with Parkinson’s disease can kill cells through ROS or other toxic substances in dopaminergic neurons [94].

PARKIN (also known as PARK2) is an E3 ubiquitin ligase recruited to OMM by PINK1 [95]. PARKIN ubiquitinates several mitochondrial proteins to stimulate mitophagy [90]. The PARKIN-mediated mitophagy is also linked to mitochondrial fission because mitochondrial fragmentation is essential for engulfment of mitochondria by autophagosomes [96]. The PINK-PARKIN pathway plays an important role in mitochondrial fusion [97,98], though the detailed mechanism remains poorly understood. Mitochondrial fission/fusion proteins are summarized in Table 1.

## 3. Cellular Metabolism and Mitochondrial Dynamics

The well-known function of mitochondria is the production of energy in the form of ATP via oxidative phosphorylation (OXPHOS), which occurs in the mitochondrial cristae [100]. Besides, the diverse functions of mitochondria are intimately related to their morphology. However, the relationships between mitochondrial dynamics and cellular metabolism are generally veiled because of the complex mechanisms involved, the involvement of multiple factors across the cellular environment, cell type variation, and differences between metabolic cues [101]. This is evident from the challenges faced by many researchers in identifying the machinery of mitochondrial dynamics and metabolism in different cell types; many pioneering studies on mitochondrial dynamics from yeast have tried to address these challenges.

### 3.1. The Mitochondrial Morphologies and Energy Metabolism in Various Stem Cells

Well-developed mitochondria are generally thought to produce energy, or ATP, more efficiently than the immature, globular mitochondria. Since well-developed mitochondria have complex cristae structures, they have a greater surface area to accommodate a larger number of inter-membrane proteins for energy production [102]. In fact, some reports showed that fused and interconnected mitochondrial structures are found in cells that depend mainly on OXPHOS for energy production [13]. However, the cells that have non-fused spherical (immature form) mitochondria have a tendency to produce energy mainly via glycolytic metabolism [103]. Therefore, cell types containing poorly developed mitochondria mainly have OXPHOS-independent metabolism. However, there are some exceptions to this, as is seen in various stem cell types (Table 2). Actively proliferating cells, such as stem cells and cancer cells, use aerobic glycolysis for energy production.

There are two types of pluripotent stem cells (PSCs), naïve and primed PSCs. Naïve PSCs, such as mouse embryonic stem cells (mESCs), are the in vitro counterparts of the inner cell mass (ICM) of preimplantation blastocyst, while primed PSCs, such as mouse epiblast stem cells (mEpiSCs), are the in vitro counterparts of epiblast of post-implantation embryos. The morphology of mEpiSC mitochondria is more tubular and has a fused shape compared to naïve pluripotent state mESCs, but mESCs have showed higher OXPHOS activity than mEpiSCs [104]. 

Moreover, embryonic mouse neural stem cells (NSCs) have been found to depend on aerobic glycolytic metabolism though they have a relatively fused mitochondrial network [106,110]. On the other hand, neurons terminally differentiated from NSCs rely on OXPHOS for energy metabolism, even if they have a fused mitochondrial network similar to NSCs. Meanwhile, mouse NPCs, whose differentiation state is in between NSCs and neurons, had more fragmented mitochondria compared with NSCs and neurons; however, they utilized aerobic glycolysis for major energy metabolism [106]. A metabolic shift from glycolysis to OXPHOS during the differentiation of NSCs occurred around the time of transition from NSCs to intermediate progenitor cells [105]. 

MSCs, which have multipotent differentiation potential to all blood cell types, have glycolysis-dependent energy metabolism [107]. They showed the relatively more tubular shape of mitochondria that can further elongate upon differentiation, as observed in NSC differentiation [111]. 

HSCs mainly use glycolysis for energy metabolism [108]. However, further differentiated HPCs were suggested to be more OXPHOS-dependent than HSCs [109]. This phenomenon might be a response of HSCs in the hypoxic environment of bone marrow to limit the production of ROS from the respiratory chain complexes in mitochondria [112]. Recently, Luchsinger et al. showed that the differentiation of HSCs was accompanied by mitochondrial OXPHOS activation. HSCs express more Mfn2 than differentiated hematopoietic lineages, indicating that they contain elongated mitochondria as the Mfn2 expression level is correlated with mitochondrial length [113]. 

### 3.2. Warburg Effect: Survival Strategy of Proliferative Glycolytic Cells

As described above, most adult stem cells mainly used aerobic glycolysis for ATP production. This kind of phenomenon found in stem cells is known as the “Warburg effect”, which was first described in cancer cells [114]. Most cancer cells produce energy through a high rate of glycolysis even when there is sufficient oxygen supply, a phenomenon termed the “Warburg effect”. The precise mechanism of the Warburg effect remains unknown. This phenomenon also came into the spotlight in the process of cellular reprogramming, or induced pluripotent stem cell (iPSC) generation [115] and a metabolic switch from OXPHOS in mouse embryonic fibroblasts (MEFs) to glycolysis in reprogrammed iPSCs. This phenomenon is commonly observed in various kinds of cancers, which display a highly proliferative state. Then, this raises the question of why proliferating cells choose an inefficient pathway to produce energy? Cell division requires not only energy, but also various kinds of cellular constituents, such as nucleotides, amino acids, and lipids. Glycolysis along with the pentose phosphate pathway can account for cellular constituents as well as ATP [116]. A reduction in mitochondrial metabolism may also allow a low level of harmful free radicals, such as ROS. Therefore, glycolysis would be beneficial to the actively proliferating stem cells to self-renew and maintain cell states [117].

### 3.3. Metabolic Regulation in Mitochondria for the Maintenance of the Pluripotent State

In addition to energy metabolism, mitochondria also play a crucial role in the stemness of PSCs. For example, PSCs showed a high level of uncoupling protein 2 (UCP2) protein [118], which is located in the membrane between the inter-membrane space and the matrix, and functions in metabolite transportation to the outside of the mitochondria, thereby regulating glucose and glutamine oxidation [119]. Mitochondrial metabolism is also important for the self-renewal capacity of human PSCs. Zhang et al. reported that although glycolysis supported the stemness of human PSCs under all conditions, oxidative mitochondrial metabolism was also highly active in human PSCs when they were cultured in a media containing lipid supplements [120]. This may highlight the importance of the cellular environment or culture condition, which could affect mitochondrial function and related mitochondrial morphology in human PSCs.

## 4. Mitochondrial Dynamics in the Reprogramming Process

The proteins related to mitochondrial dynamics, such as fusion and fission, are, interestingly, crucial for pluripotential reprogramming. During reprogramming (iPSC generation) and re-differentiation of iPSCs, mitochondrial morphology dynamically changes (Figure 2); mitochondria become elongated during reprogramming and become globular-shaped after re-differentiation into a neural lineage [121]. As the mitochondrial morphology changes dynamically during the process of reprogramming, the metabolic profile switches from OXPHOS to glycolysis [121,122,123,124]. Several studies have suggested that the mitochondrial dynamics and energy metabolism are critical for the reprogramming process.

### 4.1. Mitochondrial Fission Proteins Affect Pluripotential Reprogramming

Here, we will discuss how mitochondrial fission proteins affect the reprogramming process. Mdivi-1 treatment, which inhibits mitochondrial fission protein, DRP1, was sufficient to suppress the early stage of reprogramming of somatic cells [125]. Moreover, iPSCs lost pluripotency when exposed to Mdivi-1, indicating that mitochondrial fission is important for gaining and maintaining pluripotency. 

Reduced expression 1 (REX1), which function in the maintenance of pluripotency, induces phosphorylation of DRP1 at Ser616 and mitochondrial fission [126]. On the other hand, the inhibition of the oncogenic mitogen-activated protein kinase (MAPK) cascade leads to robust mitochondrial fusion via the loss of phosphorylation in DRP1 at Ser616 through ERK1/2 protein [127]. Therefore, Drp1 phosphorylation by the ERK pathway is necessary for the pluripotential reprogramming process [128]. Furthermore, inhibition of the accessory proteins, such as Gdap1, Mid51, and Mff, which control Drp1 recruitment to the mitochondria, suppressed reprogramming due to impaired mitochondrial fission. In particular, Gdap1-null cells displayed G2/M growth arrest in cells undergoing reprogramming and affected the early phase in reprogramming [129]. Collectively, changes in mitochondrial dynamics and the cell cycle are crucial factors for the efficient reprogramming of cells.

### 4.2. Mitochondrial Fusion Proteins Affect Pluripotential Reprogramming

Mitochondrial fusion proteins can also affect reprogramming efficiency through a different pathway from that of fission proteins. Son et al. revealed that depletion of mitochondrial fusion proteins, such as Mfn1 and Mfn2, increased the efficiency of somatic cell reprogramming into iPSCs as well as maintained pluripotency [130]. They also showed that Mfn1 and Mfn2 depletion facilitates the transition of OXPHOS to glycolytic metabolism because Mfn1 and Mfn2 are inhibitors of reprogramming as they directly bind to Ras and Raf and thus inhibit cell proliferation. Inhibition of Mfn1 and Mfn2 also activated ROS-mediated hypoxia-inducible factor 1α (HIF1α) signaling at an early stage and facilitated the reprogramming of a favorable hypoxic condition [130].

## 5. Mitochondrial Dynamics in the Differentiation Process

During the process of PSC differentiation, changes in mitochondrial morphology and metabolite composition are essential among the various differentiated cell types [131]. As PSCs mainly use glycolysis, and differentiated cells use OXPHOS for ATP production, respectively, inhibition of mitochondrial OXPHOS during the differentiation of PSCs leads to a defect in differentiation and instead, supports the maintenance of pluripotency. In line with this, the proteins related to mitochondrial dynamics also play a crucial role in the differentiation process.

### 5.1. Mitochondrial Fission Proteins Affect Cellular Differentiation

Several reports showed that Drp1-dependent mitochondrial fission is crucial for embryonic and cellular differentiation in vivo and in vitro. Drp1-null mice showed defective trophoblast giant cells and decreased cardiomyocyte beat rates and died around 11.5 dpc; however, they showed normal levels of intracellular ATP [132,133]. Conditional knockout of Drp1 showed a defect in cerebella during postnatal development. Neural cell-specific Drp1 knockout mice displayed brain hypoplasia and in vitro culture of the forebrain showed a reduction in the number of neurites and abnormal synapse formation [132,133]. However, heterozygote knockout of Drp1 did not affect mitochondrial and synaptic viability [134]. Kim et al. showed that inhibiting Drp1 activity induced a morphological change of migratory adult NSCs, which caused abnormal migration and prevention of neuronal differentiation in the NSCs [135]. 

During cellular maturation, mitochondrial localization and distribution are under the control of cellular states, such as cell division, migration, etc. Therefore, the localization of mitochondria is dynamically regulated during neuronal maturation and myogenic differentiation [136]. During neuronal differentiation, mitochondria accumulate in the regions where high energy is required, such as growth cones, at the early stage of differentiation and then localize at presynaptic terminals following neuron maturation [135]. In myogenic differentiation, NO/cGMP control Drp1 localization and activity and stimulate myogenesis through inhibition of Drp1-dependent mitochondrial fission [136]. Furthermore, Mdivi-1 mediated Drp1 inhibition suppressed expression levels of key myogenic regulatory factors (MRFs), such as MyoD and Myogenin, in differentiating C2C12 cells, a mouse muscle myoblast [137]. Likewise, myogenic differentiation of C2C12 myoblasts required Drp1-mediated mitophagy [138]. A recent report also suggested that knock-down or inhibition of Drp1 by using Mdivi-1 promotes differentiation into cardiac mesoderm lineage from human PSCs [139]. 

Fis1 function in stem cells and differentiation has only been reported recently. Pei et al. showed that the gene expression level of Fis1 was specifically high in leukemia stem cells and it functions as a crucial mitochondrial morphology regulator [140]. They also showed that loss of Fis1 impairs mitochondrial dynamics and induces myeloid differentiation in acute myeloid leukemia.

### 5.2. Mitochondrial Fusion Proteins Affect Cellular Differentiation

Besides mitochondrial fission, mitochondrial fusion additionally executes crucial roles in cellular differentiation process, especially in cardiac, neural, and mesenchymal differentiation. The deletion of Mfn1 and Mfn2 in the mouse embryonic hearts impaired mouse heart development, and ablation of Mfn2 or Opa1 in mouse ESCs resulted in defective cardiac differentiation of the ESCs [141]. Gene expression profiling showed that mitochondrial morphology-related genes interacted with calcineurin to regulate Notch1 signaling that controls cardiac differentiation [141]. Similarly, proteins that drive mitochondrial fusion, such as MFN (mitofusin) 1 and 2 and OPA1, are required for the differentiation of stem cells into cells that depend on OXPHOS metabolism, like cardiomyocytes and neurons [141,142].

Mitochondrial dynamics are also involved in MSC differentiation, including adipogenesis, osteogenesis, and chondrogenesis. Mitochondrial elongation (increase in Mfn1 and Mfn2 expression) is correlated with the adipogenesis and osteogenesis, and mitochondrial fragmentation (increased expression of Drp1, Fis1, and Fis2) is involved in chondrogenesis. Consequently, knockdown of Mfn2 and the overexpression of a dominant negative form of Drp1 resulted in defective differentiation in adipo- and osteogenesis, and chondrogenesis, respectively [111]. 

During the differentiation of human iPSCs into neurons, the expression level of Mfn2 increased with time after differentiation [142]. Knockdown of Mfn2 results in mitochondrial dysfunctions, such as downregulated expression of complexes I and IV, and ATP levels, and impaired neuronal differentiation. On the contrary, Mfn2 overexpression in NPCs promotes neuronal differentiation with enhanced mitochondrial bioenergetics and functions. Taken together, many studies have shown that mitochondrial fusion and fission play crucial roles in various cellular differentiation processes through the control of bioenergetics, signaling pathways, or expression of tissue-specific genes.

## 6. Mitochondrial Dynamics in Aging

Mitochondria also play an important role in cellular aging and cell death associated with necrosis, apoptosis, autophagy, and mitophagy through the modulation of redox by reduction reaction mechanisms [143,144]. The process of aging may be associated with the accumulation of damages, such as the production of metabolic by-products and ROS, accumulation of biological waste products, telomere shortening, and dysregulation of metabolic pathways [145,146,147]. Most of these aging factors are associated with mitochondrial dynamics and functions, indicating the close relationship between aging and mitochondria.

### 6.1. Mitochondrial ROS Impact on Cellular Senescence-Related Mitochondrial Dynamics

Abnormally elongated mitochondria are often observed in various senescent cells, implying that mitochondrial dynamics may have a functional role in cell senescence and aging. This phenomenon could be caused by the alteration of expression patterns of genes associated with mitochondrial fission, such as Drp1 and Fis1, and with mitochondrial fusion, such as Mfn1 and Mfn2. Mai et al. reported that the senescent human endothelial cells (HUVECs) showed reduced expression levels of DRP1 and FIS1 that caused long interconnected mitochondria [148]. The loss of DRP1 exacerbated endothelial cell dysfunction by inhibiting autophagic flux accompanied by increasing mitochondrial ROS [149]. In addition, the regulation of ROS by mitochondrial fission was dependent on protein disulfide isomerase A1 (PDIA1) in mouse endothelial cells; PDIA1-depleted endothelial cells activated mitochondrial fission [150]. 

Recently, Leduc-Gaudet et al. also suggested that the levels of mitochondrial dynamics-related proteins, including Mfn1, Mfn2, Opa1, and Drp1, were not significantly different between young and aged skeletal muscles [151]. However, the ratio between Mfn2 and Drp1 protein expression levels could define the extent of aging in skeletal muscle cells; as skeletal muscle cells grew older, the ratio of Mfn2/Drp1 significantly increased [151]. On the other hand, Debastian et al. suggested that the expression level of Mfn2 in skeletal muscle decreased during aging [152]. Mfn2 deficiency in skeletal muscle caused the reduction of mitochondrial respiration and elevation of oxidative stress, which was accompanied by the activation of the transcription factor, HIF1α, to minimize the accumulation of damaged mitochondria [152]. Thus, Mfn2 functions as a regulator in mitophagy and consequently controls the mitochondrial quality control pathway. 

Moreover, the stress-responsive mitochondrial protein, sirtuin 4 (Sirt4), was suggested to have implications in aging. Similar to Mfn2-functioning in mitophagy, Sirt4 could promote mitochondrial fusion by interacting with Opa1 and reducing mitophagy [153]. Overall, various aging factors are interconnected, and of these, mitochondrial dysfunction and dynamics are the underlying mechanisms of cellular aging.

### 6.2. Mitochondrial Involvement in Cell Death Activation and Calcium Signaling

Mitochondrial calcium, which is delicately controlled, functions as a cellular signal transporter, such as in cellular energy production and cell death [154]. Actually, increased mitochondrial calcium levels could induce mitochondrial fragmentation by fission proteins, Drp1 and Fis1, and it is linked to the oligomerization of Bak-induced apoptosis [27,155]. Recently, Wang et al. showed that clearance of apoptotic cells by phagocytes, a process called efferocytosis, occurred in a Drp1-dependent manner with calcium transporters [156]. Besides, calcium influx affects calcineurin-dependent Drp1 dephosphorylation, causing cell death [157]. Also, Bax/Bak-mediated induction of Drp1 causes mitochondrial fission and cell death, eventually [158]. Therefore, mitochondrial fission, calcium uptake, and cell death are closely related. 

In addition, Mfn2 could regulate calcium signaling in an ER dependent manner. For connection with mitochondria, the ER uses various proteins, such as Mfn2, VAPB (vesicle-associated membrane protein-associated protein B), and Bap31, which were connected with mitochondria-located Mfn1/2, PTPIP51 (protein tyrosine phosphatase interacting protein 51), and Fis1, respectively [159]. Mfn2 in the ER serves as the point of juxtaposition with mitochondria [160,161], and this connection enables inter-organellar calcium signaling between the two organelles. Although further extensive studies are needed, it is suggested that the proteins associated with mitochondrial dynamics dramatically affect cell death and calcium signaling.

## 7. Conclusions and Perspective

In this review, we have discussed the widespread involvement of mitochondria in various cellular processes, such as cell survival, cell cycle, proliferation, differentiation, reprogramming, aging, and energy metabolism. The variety of functions carried out by mitochondria implicates that the normal mitochondrial dynamics controlled by mitochondrial fusion/fission are critical for human health. Clinically, defects in mitochondrial fusion/fission could cause diseases associated with optic atrophy, Charcot-Marie-Tooth disease, and other neurodegenerative disorders [162,163,164,165,166,167]. There are a few studies about the relationship between mitochondrial dynamics proteins and various cellular processes. For example, we already knew that when Drp1 is knocked-out or knocked-down, mitochondrial morphology is elongated [168]. However, so far, there is a lack of understanding of energy metabolism, proliferation, and cell cycle variation caused by mitochondrial morphology changes in various cell types and conditions. Ultimately, we need to study each mitochondrial fission/fusion gene closely in relation to the various cellular process. 

Further understanding of mitochondrial dynamics is of paramount importance in elucidating mechanisms of diseases at the cellular level and discovering novel therapies to cure associated diseases. Recently, zinc finger nucleases (ZFN) and transcription activator-like effector nucleases (TALEN) were used to recover the disease-related phenotypes in mice [169,170]. Mutated mitochondrial DNA (mtDNA), which is related to skeletal and cardiac muscle, could be repaired by mitoZFN and mitoTALEN and reduce mutant mtDNA. Along with these two approaches, the CRISPR/Cas9 system will be applicable for the cure of neurological disorders caused by mutations in mitochondrial dynamic genes.

## Figures and Tables

**Figure 1 ijms-19-03893-f001:**
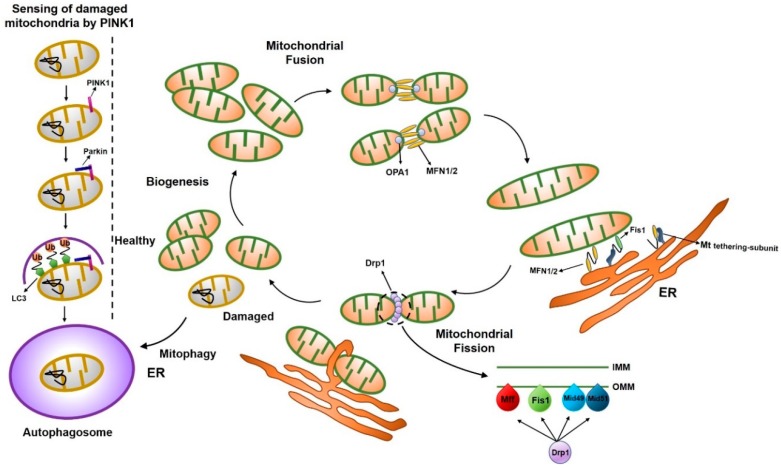
Schematic illustration of the mitochondrial dynamics. Mitochondria dynamically change their morphology through the cycle of fusion and fission. Main fusion factors are Opa1, Mfn1, and Mfn2, which bind to the inner membrane (IMM) and outer membrane (OMM) of mitochondria. Drp1 is a major fission factor that binds to OMM and forms a ring-like structure around mitochondria, leading to the separation of mitochondria into two, where endoplasmic reticulum (ER) contact occurs. Mff, Fis1, Mid49, and Mid51 function as adaptors to recruit Drp1 to the OMM.

**Figure 2 ijms-19-03893-f002:**
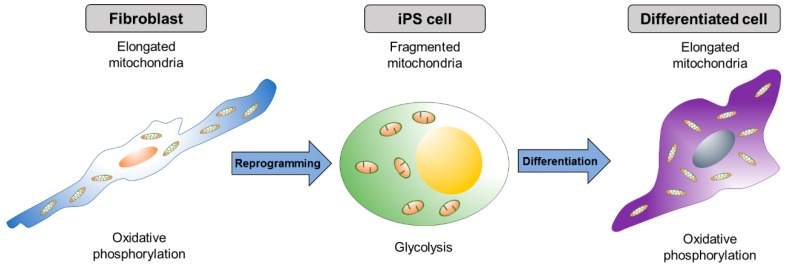
Dynamic change of mitochondrial shape during reprogramming and differentiation. Elongated mitochondria in differentiated cells become spherical shaped during the formation of iPSCs. As iPSCs differentiate, mitochondria resort back to the elongated shape.

**Table 1 ijms-19-03893-t001:** Mammalian mitochondrial fission/fusion proteins.

Component	Location	Size	Function	Reference
Drp1	OMM	80 kDa	Fission main protein	[40,99]
Fis1	OMM	18 kDa	Fission accessory protein	[43,99]
Mff	OMM	55 kDa	Fission accessory protein	[56,99]
MiD51	OMM	51 kDa	Fission accessory protein	[59,99]
MiD49	OMM	49 kDa	Fission accessory protein	[59,99]
GDAP1	OMM	16~21 kDa	Fission accessory protein	[63]
Endophilin	Intracellular membrane	40~43 kDa	Fission accessory protein	[66]
Opa1	IMM	86 kDa	Fission/Fusion protein	[21]
Mfn1	IMM	84 kDa	Fusion main protein	[19]
Mfn2	IMM	86.1 kDa	Fusion main protein	[19]
PINK1	IMM	63 kDa	Fusion accessory protein	[90]
PARKIN	IMM	52 kDa	Fusion accessory protein	[90]

**Table 2 ijms-19-03893-t002:** Mitochondrial morphology and energy metabolism in various cell types.

Cell type	Potency	Morphology	Predominant Energy Metabolism	Reference
Embryonic stem cells (ESCs)	Naïve pluripotency	Non-fused spherical	Glycolysis (Higher OXPHOS than EpiSCs)	[104]
Epiblast stem cells (EpiSCs)	Primed pluripotency	Non-fused spherical	Glycolysis	[104]
Neural stem cells (NSCs)	Multipotency	Fused elongated	Glycolysis	[105,106]
Neural progenitor cells (NPCs)	Multipotency	Non-fused spherical	Glycolysis	[105,106]
Neurons	-	Fused elongated	OXPHOS	[105,106]
Mesenchymal stem cells (MSCs)	Multipotency	Fused elongated	Glycolysis	[107]
Hematopoietic stem cells (HSCs)	Multipotency	Fused elongated	Glycolysis	[108]
Hematopoietic progenitor cells (HPCs)	Multipotency	Fused elongated	Glycolysis (Higher OXPHOS than HSCs)	[109]

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
