# Peer review of "Mitochondrial Dynamics in Stem Cells and Differentiation"

_ijms, 2018, doi:10.3390/ijms19123893_

Reviewer 1 Report
please see attached

Author Response
We appreciate your thoughtful suggestions and insights. We have revised all of your suggestions in the re-submitted manuscript. Thanks to your care, this manuscript is closer to the publication. Best wishes for a successful and rewarding year.
Reviewer 2 Report
The authors tried to prepare a comprehensive review on mitochondrial dynamics with special focus on things relevant to stem cell biology. Titles of individual sections were well chosen. Key protein components and their roles in mitochondrial dynamics are described in detail.
1_ Even though the authors said that they will describe the relevant mechanisms in the beginning of the manuscript, they finally mentioned that most mechanisms have not been clearly understood in the main body of the manuscript. So please rewrite the introduction not to mislead readers.
2_ Fission and fusion are key steps of mitochondrial dynamics, but not mitochondrial dynamics itself. The authors need to describe the time scale of fission and fusion. In addition, readers would be interested in how fission and fusion are balanced. Please add a section for this. Throughout the manuscript, much more topics and findings about DYNAMICS (=Changing events over time) need to be described and explained based on the paper title.
3_ Each section can be better connected to other sections for the manuscript to be a comprehensive review. Please do not just list the protein components and their roles in each section. Please consider the overall structure of the manuscript and interconnections among sections.
4_ They mentioned many proteins involved in fission and fusion. If they provide a table to summarize the proteins and their roles, it would be great for readers.
5_ Many descriptions are not complete.
For example,
119 Mfn1 and Mfn2 share about 80% similarity and show the same structural motifs.
The authors need to more clearly describe how similarity was defined.
43 Mitochondrial fission is regulated by post-43 translational modification of the Drp1 protein,
44 including modification by phosphorylation, S-nitrosylation, small ubiquitin-like modifier
45 (SUMO)-ylation, ubiquitination, and O-GlcNAc modification (O-GlcNAcylation) in response to a
46 variety of cellular stimuli [12].
The authors need to explain how each version of modification affects mitochondrial fission.
227 ROS. Therefore, glycolysis would be beneficial to the actively proliferating stem cells to self-renew
228 and maintain cell states [97].
To be more quantitative, the portion of use of glycolysis compared with alternative or parallel energy metabolism pathways need to be described. In addition, the authors need to refer to more works to reach such a conclusion.
6_ There are so many repeated phrases and sentences throughout the manuscript. Please carefully revise them.
7_ Paper citation should be done in the numerical order.
Author Response
The authors tried to prepare a comprehensive review on mitochondrial dynamics with special focus on things relevant to stem cell biology. Titles of individual sections were well chosen. Key protein components and their roles in mitochondrial dynamics are described in detail.
1_ Even though the authors said that they will describe the relevant mechanisms in the beginning of the manuscript, they finally mentioned that most mechanisms have not been clearly understood in the main body of the manuscript. So please rewrite the introduction not to mislead readers.
>> We have rewritten the last section of the introduction.
“In this review, we describe the basic mechanisms of functioning of the proteins involved in mitochondrial dynamics, without details on the relative mechanisms.”
2_ Fission and fusion are key steps of mitochondrial dynamics, but not mitochondrial dynamics itself. The authors need to describe the time scale of fission and fusion. In addition, readers would be interested in how fission and fusion are balanced. Please add a section for this. Throughout the manuscript, much more topics and findings about DYNAMICS (=Changing events over time) need to be described and explained based on the paper title.
>> We have added a relevant paragraph in the manuscript (2. Components determining the mitochondrial structure, line 43-50).
3_ Each section can be better connected to other sections for the manuscript to be a comprehensive review. Please do not just list the protein components and their roles in each section. Please consider the overall structure of the manuscript and interconnections among sections.
>> We agree with your opinion. The effect of mitochondrial fission/fusion proteins directly related to the mitochondrial dynamics has been described. However, details of these proteins in the other functions such as metabolism, reprogramming, differentiation, and aging is poorly clarified. In addition, few articles cover the full story of each gene. So, we have decided to list and explain the proteins revealed in each section.
4_ They mentioned many proteins involved in fission and fusion. If they provide a table to summarize the proteins and their roles, it would be great for readers.
>> We have added a table (Table 1. Mitochondrial fission/fusion proteins in mammals) in the manuscript.
5_ Many descriptions are not complete.
For example,
119 Mfn1 and Mfn2 share about 80% similarity and show the same structural motifs.
The authors need to more clearly describe how similarity was defined.
>> We have added more information about the similarity.
43 Mitochondrial fission is regulated by post-translational modification of the Drp1 protein,
44 including modification by phosphorylation, S-nitrosylation, small ubiquitin-like modifier
45 (SUMO)-ylation, ubiquitination, and O-GlcNAc modification (O-GlcNAcylation) in response to a
46 variety of cellular stimuli [12].
The authors need to explain how each version of modification affects mitochondrial fission.
>> We have added more information about this issue in the main text and references.
227 ROS. Therefore, glycolysis would be beneficial to the actively proliferating stem cells to self-renew
228 and maintain cell states [97].
To be more quantitative, the portion of use of glycolysis compared with alternative or parallel energy metabolism pathways need to be described. In addition, the authors need to refer to more works to reach such a conclusion.
>> We tried to find the related paper, but we could not find any research papers describing the quantitative ratio of glycolysis and OXPHOS in stem cells. So, we have used an ambiguous expression such as ‘beneficial’. In references numbered 116, 117, some clues regarding the ‘beneficial’ reasons for glycolysis in stem cells were described.
6_ There are so many repeated phrases and sentences throughout the manuscript. Please carefully revise them.
>>We have checked repeated phrases and sentences and revised them.
7_ Paper citation should be done in the numerical order.
>> We have rewritten the citations in numerical order.
Reviewer 3 Report
In this review article, the authors reviewed the literature related to mitochondrial dynamics and associated processes in stem cells pluripotency and differentiation. The authors focused on the various proteins involved in mitochondrial fission and fusion, mitochondrial dynamics and metabolism, mitochondrial dynamics in reprogramming, differentiation and aging. This is an exciting and timely review article. The authors have covered various aspects of mitochondrial dynamics. The manuscript is very well written and presented in a logical manner. However, this review article has some weaknesses, which needs to be addressed during revision.
Specific Comments:
1. Most of the discussed topics in the review describe what is known, but critical inputs are not provided. The author may discuss the known fact for each topic, then they should provide their own perspectives on each topic, how future research may further advance this area of research.
2. Although the authors have covered multiple aspects of the mitochondrial dynamics. The structural homeostasis of the mitochondrial network is the outcome of multiple dynamics related processed like, biogenesis, fission-fusion, shape remodeling, and mitophagy. It would be great to incorporate these individual signaling processes in the context of pluripotency and differentiation.
3. Recent evidence clearly suggests the imperative role of ER-mitochondrial contacts in the regulation of mitochondrial dynamics. Please discuss this aspect in the context of stem cells and differentiation along with detailed mechanism.
4. The authors should provide more in-depth information on each of the discussed topics in the review. Some aspects are completely ignored which needs to included. For example, mitochondrial involvement in cell death activation during aging, the role of mitochondrial protein homeostasis and calcium signaling in mitochondrial dynamics and stem cells.
5. Numerous spelling mistakes throughout the manuscript. The manuscript needs a thorough revision in terms of typos and grammar.
6. Please correct the MOM to OMM as outer mitochondrial membrane, MIM to IMM as inner mitochondrial membrane.
7. Figure 1: incorporate all the molecular processes involved in mitochondrial dynamics, i.e., biogenesis, fission-fusion, structural remodeling, mitophagy and role of inter-organelle crosstalk in the maintenance of mitochondrial dynamics.
Author Response
In this review article, the authors reviewed the literature related to mitochondrial dynamics and associated processes in stem cells pluripotency and differentiation. The authors focused on the various proteins involved in mitochondrial fission and fusion, mitochondrial dynamics and metabolism, mitochondrial dynamics in reprogramming, differentiation and aging. This is an exciting and timely review article. The authors have covered various aspects of mitochondrial dynamics. The manuscript is very well written and presented in a logical manner. However, this review article has some weaknesses, which needs to be addressed during revision.
Specific Comments:
1. Most of the discussed topics in the review describe what is known, but critical inputs are not provided. The author may discuss the known fact for each topic, then they should provide their own perspectives on each topic, how future research may further advance this area of research.
>> We have added our perspectives and future directions in the Discussion section.
2. Although the authors have covered multiple aspects of the mitochondrial dynamics. The structural homeostasis of the mitochondrial network is the outcome of multiple dynamics related processed like, biogenesis, fission-fusion, shape remodeling, and mitophagy. It would be great to incorporate these individual signaling processes in the context of pluripotency and differentiation.
>> We have explained the relationship between the Drp1 protein and the ERK pathway in the introduction of the pluripotency paragraph (4.1). However, we could not find papers dealing with signaling, which induces differentiation and affects mitochondrial fission / fusion proteins.
3. Recent evidence clearly suggests the imperative role of ER-mitochondrial contacts in the regulation of mitochondrial dynamics. Please discuss this aspect in the context of stem cells and differentiation along with detailed mechanism.
>> We have added ER-mitochondrial contacts in regulation of mitochondrial dynamics in section 2.2, Mitochondrial fusion proteins.
4. The authors should provide more in-depth information on each of the discussed topics in the review. Some aspects are completely ignored which needs to be included. For example, mitochondrial involvement in cell death activation during aging, the role of mitochondrial protein homeostasis and calcium signaling in mitochondrial dynamics and stem cells.
>> We have added a relevant paragraph in the manuscript (6.2. Mitochondrial involvement in cell death activation and calcium signaling)
5. Numerous spelling mistakes throughout the manuscript. The manuscript needs a thorough revision in terms of typos and grammar.
>> We have checked the manuscript and revised them.
6. Please correct the MOM to OMM as outer mitochondrial membrane, MIM to IMM as inner mitochondrial membrane.
>> We have corrected MOM to OMM and MIM to IMM in the manuscript
7. Figure 1: incorporate all the molecular processes involved in mitochondrial dynamics, i.e., biogenesis, fission-fusion, structural remodeling, mitophagy and role of inter-organelle crosstalk in the maintenance of mitochondrial dynamics.
>> We have revised figure 1 following your suggestions.
Round 2
Reviewer 2 Report
None
Author Response
we have modified several points from the academic editor
Response to academic editor’s comments
The peer review process for your revised manuscript, entitled "Mitochondrial dynamics in stem cells and differentiation”, is now complete. It has been determined that this manuscript will be acceptable for publication in IJMS pending some minor revisions.
1. Figure 1 – The statement on lines 197/198 that mitochondrial fragmentation is essential for engulfment of mitochondria by autophagosomes, is not reflected in Figure 1. In this figure, an elongated mitochondrion is sequestrated within the autophagosome (and, it is well-known that elongated mitochondria are spared from autophagic degradation [Gomes et al. (2011) Nat Cell Biol 13, 589-598]). Please modify the figure accordingly.
à We have modified Figure 1.
2. Lines 72 and 118. Please rephrase the sentences without “recently”. The references cited date already from 2009 and 2010, respectively. Note that the same is true for line 104. However, at this position, please also refer to “Friedman et al. (2011) Science 334, 358-362”, who reported the phenomenon under review for the first time.
à We have rephrased the sentence on Line 72, 104, and 118, and also referred the paper by Friedman et al. published in Science.
3. Table 1. – The statement that Fis1 is the “Fission main protein” and Mff is the “Fission accessory protein” is controversial, and even questionable (see [Osellame et al. (2016) J Cell Sci 129, 2170-2181); and references therein). Please modify accordingly.
à We have modified this point.
4. Line 410: “… by fusion proteins Drp1 and Fis1”. Drp1 and Fis1 are protein involved in fission (and not fusion).
à We have corrected this point.
Reviewer 3 Report
The authors addressed all the reviewer concerns. The manuscript can be accepted.
Author Response
we have modified several points from the academic editor